# Accuracy of Inertial Measurement Units When Applied to the Countermovement Jump of Track and Field Athletes

**DOI:** 10.3390/s22197186

**Published:** 2022-09-22

**Authors:** Paulo Miranda-Oliveira, Marco Branco, Orlando Fernandes

**Affiliations:** 1Interdisciplinary Research Centre Egas Moniz (CiiEM), Cooperativa de Ensino Superior CRL, 2829-511 Almada, Portugal; 2School of Technology and Management (ESTG), Polytechnic of Leiria, 2411-901 Leiria, Portugal; 3Portuguese Athletics Federation (FPA), 2799-538 Oeiras, Portugal; 4Escola Superior de Desporto de Rio Maior, Instituto Politécnico de Santarém, 2040-413 Rio Maior, Portugal; 5Centro Interdisciplinar de Estudo da Performance Humana (CIPER), Faculdade Motricidade Humana da Universidade de Lisboa, 1495-751 Oeiras, Portugal; 6Sport and Health Department, School of Health and Human Development, Universidad de Évora, 7000-671 Évora, Portugal; 7Comprehensive Health Research Center (CHRC), University of Évora, 7000-671 Évora, Portugal

**Keywords:** IMU, CMJ, high-level athletes, contraction phase, modified reactive strength index (RSImod)

## Abstract

In this study, we aimed to assess the countermovement jump (CMJ) using a developed instrument encompassing an off-the-shelf Inertial Measurement Unit (IMU) in order to analyze performance during the contraction phase, as well as to determine the jump height and the modified reactive strength index (RSImod), using force plate (FP) data as reference. Eight athletes (six males and two females) performed CMJs with the IMU placed on their fifth lumbar vertebra. Accuracy was measured through mean error (standard deviation), correlation, and comparison tests. The results indicated high accuracy, high correlation (r), and no statistical differences between the IMU and the FP for contraction time (r = 0.902; ρ < 0.001), negative impulse phase time (r = 0.773; ρ < 0.001), flight time (r = 0.737; ρ < 0.001), jump time (r = 0.708; ρ < 0.001), RSImod (r = 0.725; ρ < 0.001), nor minimum force (r = 0.758; ρ < 0.001). However, the values related to the positive impulse phase did not have the expected accuracy, as we used different devices and positions. Our results demonstrated that our developed instrument could be utilized to identify the contraction phase, jump height, RSImod, and minimum force in the negative impulse phase with high accuracy, obtaining a signal similar to that of an FP. This information can help coaches and athletes with training monitoring and control, as the device has simpler applicability making it more systematic.

## 1. Introduction

Athletic events require specific strength and power characteristics [1]. Vertical jump (VJ) tests, such as countermovement jumps (CMJs), are simple, highly practical, and effective tests to monitor power output [1,2]; furthermore, they can assess neuromuscular function and fatigue [3], using cost- and time-effective technology [4].

There exist several instruments that can quantify CMJ parameters. A force plate (FP) is often used to measure CMJ ability, due to its high accuracy and validity, allowing parameters such as force, impulse, power, acceleration, velocity, and displacement to be determined [5]. However, FPs require extensive data processing, are expensive, and are difficult to transport [5,6,7,8,9,10,11,12,13,14]. Other validated and applied devices are more suitable, affordable, and accessible, such as jump reach systems (Vertec) [15], contact belt systems [8], the mobile phone MyJump application [16], and optical time systems (Optojump; Microgate, Bolzano, Italy) [10].

Inertial Measurement Units (IMUs) have recently been applied in VJ testing, due to their applicability on any surface, ease of transportation provided by their small size, and ability to provide temporal and kinematic parameters, which may have a direct relation to kinetic parameters [13,14,17,18,19,20]. IMUs are micro-electromechanical sensor systems (MEMSs) that incorporate an accelerometer, a gyroscope, and a magnetometer [19]. Studies have suggested many kinds of IMUs as valid devices for obtaining the jump height [11,12,13,14,17] and accuracy in order to determine the flight time and the center of mass (CoM) displacement in CMJs and CMJs with arms when applied to specific algorithms [18]. Furthermore, different IMU positions have been tested [21] in the CMJ. Rantalainen et al. [22] suggested a position on the back midline at the L3 to L5 level, and most studies placed the IMU on the trunk at the level of the fifth lumbar vertebra (L5) [11,12,13,14,18,19]; it was also placed on the calcaneus [19].

Several methods are used to determine various parameters of the CMJ. One of the most-used methods is the one proposed by Bosco et al. [7], which uses the jump flight time to analyze the explosive force through jump height [7,15]. However, Beattie et al. [4] have shown the importance of determining the contraction time (e.g., identifying the braking and propulsion phases) and analyzing the force, velocity, power, and impulse parameters when characterizing world-class male sprinters. Indeed, world-class male sprinters produced a significantly larger impulse, peak velocity, and power in both the braking and propulsion phases, as well as higher jump height and modified reactive strength index (RSImod) in the CMJ test, than sub-elite male sprinters [4].

Many studies focused on the validation of IMUs in the CMJ can be found in the literature [11,12,13,14,17,18]. Still, most of these studies explored parameters such as jump height or CoM trajectory [11,12,13,14,17,18], while few studied and analyzed the data signals [23], mainly in the CMJ contraction phase, which was suggested to be a crucial phase when analyzing the performance of high-level athletes [4]. In the present study, we aim to assess the CMJ using a developed instrument that encompasses an off-the-shelf IMU in order to analyze performance during the contraction phase, as well as to determine the jump height and RSImod, considering FP data as reference. We hypothesize that temporal parameters with a lower mean error and a similar performance may be obtained, but with different amplitudes when comparing IMU with FP data, as the devices and their positions are distinct. Additionally, we hypothesize that the parameters do not show statistically significant differences, when comparing the results from the IMU and the FP.

## 2. Materials and Methods

### 2.1. Experimental Overview

The experiment conducted in the present study consisted of a session of training monitoring and control, in which we evaluated three to five CMJs with vertical force calculated using the IMU ($Fz_{IMU}$) for comparison with the vertical force measured using a force plate ($Fz_{FP\;})$.

### 2.2. Participants

A total of 30 healthy, high-level athletes (15 males and 15 females) with international representation of their countries and without injuries in past 6 months were invited. Eight athletes—six males (four pole vaulters, 5.53 ± 0.13 m; two decathletes, 7280 ± 183 points; age, 24.33 ± 2.71; body mass (BM), 77.90 ± 8.69 Kg; and height (h), 1.80 ± 0.07 m) and two females (pole vaulters, 4.47 ± 0.04 m; age, 30.00 ± 2.00; BM, 63.23 ± 1.87 Kg; and h, 1.70 ± 0.01 m)—accepted and were included in the study. All athletes were informed about the study objectives and signed the protocol approved by the Ethical Council of Universidade de Évora (GD/46951/2019). This declaration was made in accordance with the ethical rights of the Helsinki Declaration.

### 2.3. Experimental Procedures

The athletes were informed about the objectives and performed a warm-up as indicated by their coaches in order to prepare for the training monitoring and control session. The investigator explained the protocol to all athletes. Before initiating the task, the IMU was calibrated. The athlete stood still for about 30 s, in order to obtain the base acceleration, and performed a previous jump to set initial criteria for both instruments. The athlete began from the static upright position and performed a knee flexion at 90° followed by a jump, finalizing in the initial position with arms placed at the waist. The jump was performed on an FP with the IMU placed at L5 as a reference for the CoM [11,12,13,14,18,19]. The instruments were fixed on the skin at L5 using adhesive tape (see Figure 1). The data were automatically synchronized through a Python routine, allowing us to simultaneously collect all data in the same file. Each coach defined the number of jumps performed by their athlete (between three and five CMJs), with a minute of rest between jumps, performing a total of 30 CMJs.

### 2.4. Instruments

In this study, we used an instrument developed by the investigation group, which encompasses an off-the-shelf IMU (size, 42 mm × 32 mm × 17 mm; weight, 22 g) consisting of a three-dimensional (3D) accelerometer (±16 g), a 3D gyroscope (±2000 dps), and a 3D magnetometer (±4900 µT) which collected 300 data per second (300 Hz); see Figure 2. A Bertec FP (Columbus, OH, USA) of 1.2 m × 0.6 m collecting at 1000 Hz was used as a gold standard. The FP data were collected using Qualisys Track Manager (Qualisys AB, Gothenburg, Sweden) and the IMU data using the Spyder package of Python 3.7. The communication between the IMU and the laptop was conducted using wireless communication (Wi-Fi) in order to improve the connection quality.

### 2.5. Data Processing

Data collection was carried out through Spyder 3.3.3 (Python Project Contributors). Analyses were performed using Scylab 6.0.1 (ESI Group, Paris, France). Only the vertical axis in both instruments was considered.

The vertical force ($Fz_{IMU}$, in Newtons) was calculated based on Newton’s Second Law using Equation (1):(1)$$Fz_{IMU}=Accz_{IMU}\times BM+BM$$
where $Accz_{IMU}$ represents the raw vertical acceleration of the IMU, which considers gravitational acceleration, and BM is the Body Mass. During the contraction phase, the force data from the IMU were smoothed using a Butterworth low-pass filter, and the cut-off frequency was determined using spectral power analysis [13,24]. A cut-off frequency of 20 Hz was applied. We did not apply smoothing during the flight phase because a different signal behavior was observed between the devices, as the IMU was always in contact with the athlete, while we did not have any signal from the FP during this phase.

The contraction phase and flight phase were defined manually. The contraction phase started when the body weight horizontal line crossed the vertical force signal and finished in the first minimum peak (Figure 3, Contraction Phase). The flight phase started at the first minimum peak and finished at the first maximum peak (Figure 3, Flight Phase). From this, the contraction time duration (*CT*) and flight time duration (*FT*) were determined, and the jump height (*JH*, in meters) was determined using Equation (2) [7]:(2)$$JH=\frac{9.81}{8}\times FT^{2}$$

The modified reactive strength index (*RSImod*, in meters per second) was determined through Equation (3) [4]:(3)RSImod=JH (m)/CT (s)

The contraction phase was divided manually into positive (Figure 3, Positive Impulse Phase) and negative (Figure 3, Negative Impulse Phase) impulse phases, also known as breaking and propulsion phases, respectively [25]. We determined the duration time for each phase, the minimum force during the negative impulse phase, and the maximum force during the positive impulse phase. When associated with the CMJ motion, the negative impulse phase is considered to describe the jump descendent motion (i.e., when the athlete bends the knee until 90°), while the positive impulse phase is associated with the ascendant phase of the jump (i.e., from 90° until the feet leave the ground) [25].

Each athlete’s signal was normalized to 100%, in order to consistently describe and analyze the force data signals.

### 2.6. Statistical Analysis

All jumps performed by athletes were considered in the analysis. Descriptive statistics were calculated to describe the spatiotemporal, kinematic, and kinetics data, with means and standard deviation values. Normality was verified using the Shapiro–Wilk test (*p* ≤ 0.05) [9,14,17]. An independent samples *t*-test was conducted. When normality was violated, the Mann–Whitney test was used to compare the spatiotemporal, kinematic, and kinetics data between IMU and FP values. Accuracy was determined through mean error, absolute mean error, standard deviation error, and correlation [18,26]. Pearson and Spearman correlations were determined to analyze whether the parameters had the same tendency between instruments. The correlation values suggested by Hopkins et al. [27] were considered: r ≤ 0.3, small; r between 0.3 and 0.5, moderate; and r > 0.5, high [28,29]. All statistical analyses were carried out using Jamovi software (Version 1.6; Sydney, Australia).

The quality of the signal was quantified by comparing the IMU data with the FP data, considering the variance accounted for (*VAF*), calculated using Equation (4) [30]:(4)$$VAF=100\times \left[1-\sum \;{\left(Fz_{IMU\;}-Fz_{FP\;}\right)}^{2}\;/\;\sum \;{\left(Fz_{IMU\;}\right)}^{2}\;\right]$$

## 3. Results

### 3.1. Descriptive Analysis

Table 1 provides the results of the descriptive statistical analyses for the analyzed parameters of the IMU and FP. Regarding temporal parameters, the mean contraction times (mean (SD)) of 0.761 (0.081) and 0.745 (0.083) s were calculated for the IMU and FP, respectively. The mean result of positive impulse phase time for the IMU was 0.387 (0.078) s, while that for the FP was 0.423 (0.039) s. The mean negative impulse phase time was 0.373 (0.115) s for the IMU and 0.322 (0.053) s for the FP. The mean flight time was 0.660 (0.042) s for the IMU and 0.665 (0.040) s for the FP. Mean jump heights of 0.535 (0.067) and 0.544 (0.065) m were determined for the IMU and FP, respectively. RSImod was 0.710 (0.113) m/s for the IMU and 0.738 (0.019) m/s for the FP. The maximum force from the IMU data was 1753 (270) N, while that for the FP was 2033 (262) N. The minimum force obtained was 120 (121) N for the IMU and 108 (121) N for the FP.

### 3.2. Force Data Signal

We applied VAF to compare the signal between the IMU at the L5 position and the FP for each jump and obtained a mean value of 64.75% for all athletes (Figure 4).

### 3.3. Accuracy Data

Table 2 summarizes the results of the accuracy analysis between the IMU and FP, with respect to the analyzed parameters. The contraction time obtained a mean error (SD) of −0.016 (0.045) s and an absolute mean error of 0.036 s, with a correlation (r) of 0.902 (*p* < 0.001). When comparing the results between the IMU and FP, no significant differences were observed (t = 399, *p* = 0.458). The positive impulse phase time obtained a mean error (SD) of 0.035 (0.082) s and an absolute mean error of 0.055 s, with an association of 0.230 (*p* = 0.221). When comparing the results between the IMU and FP, no significant differences were found (t = 334, *p* = 0.088). The negative impulse phase time obtained a mean error (SD) of −0.051 (0.074) s and an absolute mean error of 0.065 s, with a correlation of 0.773 (*p* < 0.001). When we compared the results between the IMU and FP, no significant differences were found (t = 324, *p* = 0.063). The flight time and jump height obtained mean errors (SDs) of −0.006 (0.030) s and 0.009 (0.049) m, absolute mean errors of 0.024 s and 0.040 m, and a correlation of 0.737 (*p* < 0.001) and 0.708 (*p* < 0.001), respectively. When comparing the results between the IMU and FP for both parameters, no significant differences were observed (t = 0.529, *p* = 0.599; t = 0.526, *p* = 0.601). RSImod obtained a mean error (SD) of −0.028 (0.095) m/s and an absolute mean error of 0.077 m/s, with a correlation of 0.727 (*p* < 0.001). When comparing the results between the IMU and FP, no significant differences were found (t = 406, *p* = 0.523). The maximum force obtained a mean error (SD) of 280 (264) N and an absolute mean error of 331 N, with a correlation of 0.491 (*p* = 0.006). When comparing the results between the IMU and FP, significant differences were found (t = 4.077, *p <* 0.001). The minimum force obtained a mean error (SD) of −13 (96) N and an absolute mean error of 76 N, with an association of 0.758 (*p* < 0.001). When comparing the results between the IMU and FP, no significant differences were found (t = 338, *p* = 0.099).

Figure 5 shows the correlations of the analyzed parameters between the IMU and the FP for the CMJ.

## 4. Discussion

In this study, we assessed the CMJ using a developed instrument that encompasses an off-the-shelf IMU in order to analyze performance during the contraction phase and determine the jump height and RSImod, considering FP data as reference. We hypothesized that temporal parameters with a lower mean error and similar performance would be obtained, but with different amplitudes when comparing IMU and FP data, as the devices and their positions were distinct.

### 4.1. Force Data Signal

Previous studies have not characterized the force data signal obtained directly with an IMU and compared them with those obtained from an FP. The results of this study indicate that our developed instrument, which encompasses an off-the-shelf IMU, could obtain behavior similar to that of an FP. Furthermore, the VAF values indicated a partial approximation between the two signals (Figure 4) [31], such that it was possible to apply the IMU to evaluate the CMJ. Moorhouse et al. [30] obtained similar results, showing no significant influence and VAF values between 74 and 80%.

Through a qualitative analysis, the performance was found to be similar when comparing IMU and FP data in the contraction phase. However, the flight phase was not similar, as expected, because there was no contact with the platform, while the IMU was always connected to the athlete. Regarding the signal structure, our study followed that of Chiang et al. [23] (p. 82), with maximum peaks before and after the minimum peaks. However, the flight phase criteria were not equal; the authors indicated zero acceleration, while we indicated a phase between the minimum and maximum peak (Figure 3, Flight Phase). Markovic et al. [19] (p. 4) showed a different way to characterize the flight time, using the maximum peaks of the absolute acceleration.

### 4.2. Accuracy of Temporal Parameters

Our developed system encompassing an off-the-shelf IMU for estimating contraction time, positive and negative impulse phase time, and flight time showed high accuracy [18]. The mean errors of contact time, positive and negative impulse phase time, and flight time were −0.016 (0.045) s, 0.035 (0.082) s, −0.051 (0.074) s, and 0.006 (0.030) s, respectively. Fathian et al. [18] (p. 10) obtained similar mean errors in measuring the flight time, of 0.03 (0.04) s, and considering their sacrum-mounted IMU to be capable of detecting jump phases with high accuracy.

Regarding the correlation analysis, the flight time had a higher correlation and comparable values [17]; the same tendency was verified for the negative impulse phase time and contraction time. However, the positive impulse phase time had a smaller correlation, but had comparable values. These results indicate that care must be taken in the analysis of the positive impulse phase, when comparing the instruments analyzed.

Information about the flight time is important, as most of the devices applied in training use the flight time to determine the jump height (Equation (2)) [8,10,13], making our device comparable with the majority of devices applied for the physical evaluation of the athletes. The positive and negative impulse phase is the differentiating factor for most devices applied in training. These results make the IMU closest to the FP, as athletes with high levels of elastic–explosive strength have superior eccentric characteristics to improve their concentric impulse, take-off velocity, and jump height [4]; therefore, it is essential to determine ways to quantify these different phases.

### 4.3. Kinematic Parameter Accuracy

The developed instrument also obtained high accuracy for jump height (0.009 (0.049) m) and RSImod (0.028 (0.095) m/s), with high correlation and comparable values, respectively. Previous studies showed the same tendency for jump height (r = 0.82–0.87) [14,18] and for RSImod, when also compared with a force plate (r = 1.00) [5]. These results reinforce that low-cost IMUs allow high-level athletes to be evaluated with high accuracy and can obtain more information than jump reach systems (Vertec), contact belt systems, the mobile phone MyJump application, and optical time systems (Optojump; Microgate, Bolzano, Italy). Regarding the obtained results, the jump height values obtained were slightly higher (Table 1) than those obtained for sub-elite male sprinters in the study by Beattie et al. [4] (0.44 ± 0.01 m), the elite sprinters in the study by Philpott et al. [1] (M, 0.464 ± 0.061 m; F, 0.371 ± 0.049 m) [1], and other studies on the CMJ [8,10,12,14,15,16,32,33], as was expected. However, our study had lower results when compared with the world-class elite male sprinters in the study by Beattie et al. [4]. These results were expected, as our study included pole vaulters and decathletes, not sprinters, and we also aggregated the results of male and female athletes, not just male athletes. Furthermore, world-class elite male sprinters require a high elastic–explosive capacity to achieve world-class performance. The high accuracy obtained for RSImod is important, as few studies refer to this parameter, but it is an important parameter for both our study and high-performance coaches, as it provides information about the stretch-shortening cycle [4] and helps to provide a detailed athlete force–time profile [5]. Athletes with high RSImod in the CMJ achieve high jump height, force values, power values, velocity values, and impulse values during the braking and propulsion phases and require a short time to take off [30]. Indeed, world-class elite male sprinters (0.83 ± 0.07 m/s) produced higher RSImod values than sub-elite athletes (0.72 ± 0.07 m/s) [4]. Our study verified smaller RSImod values than elite athletes, but with RSImod values similar to those of sub-elite athletes (Table 1) and higher values than those of high-school athletes [5]. Considering that we had high jump height and small RSImod, we can suggest that our high-level athletes need to optimize the component time to improve their impulse, a determinant parameter for improving the elastic–explosive characteristic [34,35].

### 4.4. Kinetic Parameter Accuracy

The developed device allowed us to obtain high-accuracy minimum force values (−13 (96) N) with high correlation and similar values. This can improve the information shared with the coach as, when we associate time (contraction time and negative impulse phase time) with force (minimum force), we can determine the impulse [4]. However, the maximum force results did not obtain the same accuracy (Table 2). As initially expected, the IMU position was different from that of the FP, and the mechanical concepts led us to expect higher ground reaction forces than those in the position that represents the CoM.

### 4.5. Recommendations and Practical Applications

From an engineering perspective, this study suggests the applying of a direct and simple data processing method that allows various parameters to be collected in real-time. From the coach’s perspective, this is a low-cost device that is easy to apply and transport [13]. It provides a real-time signal similar to that of an FP, with high accuracy for RSImod, jump height, contraction time, and negative impulse phase data. Additionally, we presented an instrument with similar price to a jump reach system (Vertec), contact belt systems, the mobile phone MyJump application, or an optical time system (Optojump; Microgate, Bolzano, Italy), but which provides more accurate information. All of these points are important, as the recent literature shows the importance of determining the jump height, contraction time, and RSImod for the characterization of high-level athletes [4,5]. Furthermore, this study can help to consolidate studies about the CMJ and machine learning [23].

## 5. Conclusions

In this study, we showed that an instrument based on an off-the-shelf IMU can serve as a bridge to identify the contraction phase, jump height, RSImod, and minimum force in the negative impulse phase with high accuracy, obtaining a signal similar to that of a force plate. All this information can help coaches and athletes with training monitor and control, as the device has simpler applicability making it more systematic. However, future approaches should consider the development of velocity and position algorithms, which remain the most challenging aspects for those working in biomechanics. Furthermore, there are still a gap and opportunity in the development of sports analysis with IMUs, namely, at the level of including the anteroposterior and mediolateral components of the data signal. Furthermore, IMUs may be applied for other movements, in order to analyze the other acceleration components, such as those mentioned above. Researchers should also focus on improving the communication between technologies and the training environment and on developing user-friendly interfaces for the coach/athlete.

## Figures and Tables

**Figure 1 sensors-22-07186-f001:**
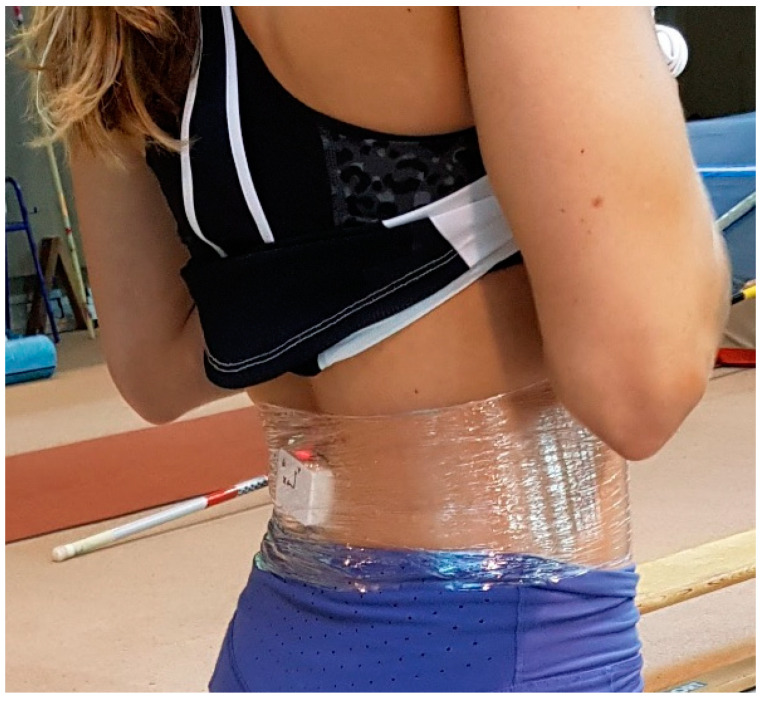
Example of the IMU location on L5.

**Figure 2 sensors-22-07186-f002:**
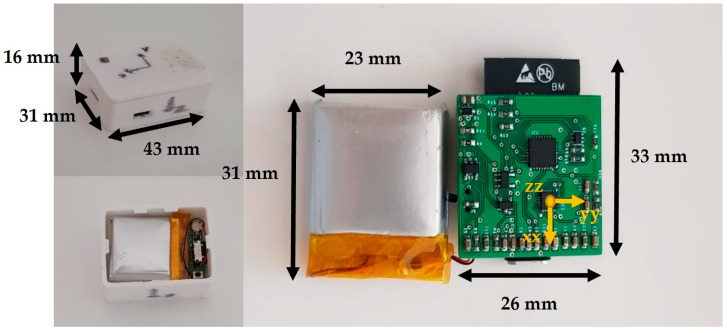
The instrument developed by the investigation group.

**Figure 3 sensors-22-07186-f003:**
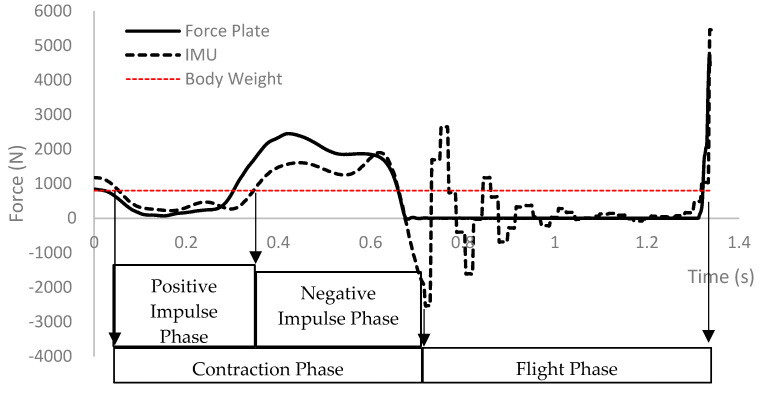
CMJ phases: contraction phase, flight phase, and positive and negative impulse phases.

**Figure 4 sensors-22-07186-f004:**
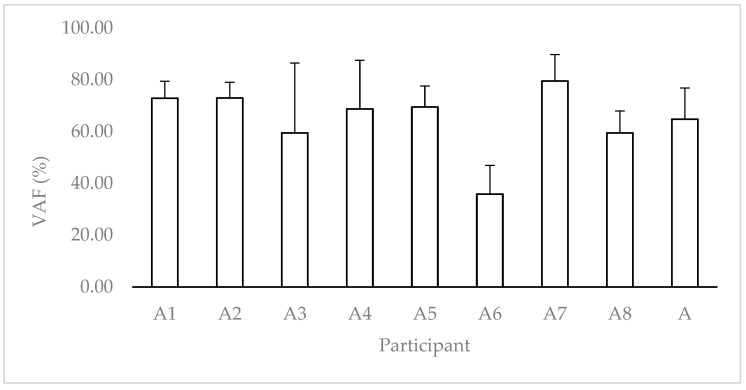
Variance accounted for, for each participant (A1–A8). A is the mean value for all participants.

**Figure 5 sensors-22-07186-f005:**
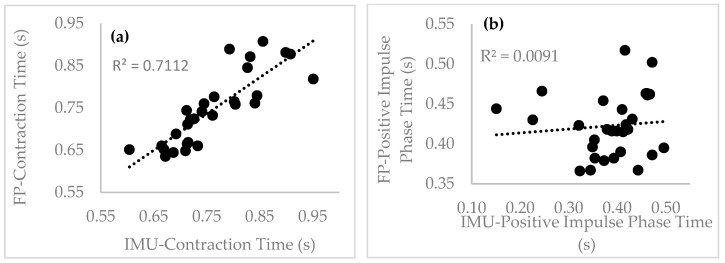
IMU vs. force plate correlation analysis: (**a**) contraction time (s); (**b**) positive impulse phase time (s); (**c**) negative impulse phase time (s); (**d**) flight time (s); (**e**) jump height (m); (**f**) RSImod (m/s); (**g**) maximum force (N); and (**h**) minimum force (N).

**Table 1 sensors-22-07186-t001:** Descriptive analysis between Inertial Measurement Unit (IMU) and force plate (FP).

Analyzed Parameter	Mean (SD)	Minimum	Maximum	
Contraction time IMU (s)	0.761 (0.081)	0.604	0.951	
Contraction time FP (s)	0.745 (0.083)	0.635	0.907	*
Positive impulse phase time IMU (s)	0.387 (0.078)	0.151	0.496	*
Positive impulse phase time FP (s)	0.423 (0.039)	0.366	0.517	
Negative impulse phase time IMU (s)	0.373 (0.115)	0.211	0.642	*
Negative impulse phase time FP (s)	0.322 (0.053)	0.262	0.445	*
Flight time IMU (s)	0.660 (0.042)	0.532	0,734	
Flight time FP (s)	0.665 (0.040)	0.562	0.734	
Jump height IMU (m)	0.535 (0.067)	0.347	0.660	
Jump height FP (m)	0.544 (0.065)	0.387	0.660	
RSI modified IMU (m/s)	0.710 (0.113)	0.505	1.090	*
RSI modified FP (m/s)	0.738 (0.119)	0.567	1.020	*
Maximum force IMU (N)	1753 (270)	1211	2291	
Maximum force FP (N)	2033 (262)	1516	2485	
Minimum force IMU (N)	120 (121)	−179	386	
Minimum force FP (N)	108 (121)	−0.746	562	*

**p* ≤ 0.05, non-normal distribution.

**Table 2 sensors-22-07186-t002:** Accuracy analysis between Inertial Measurement Unit (IMU) and force plate (FP).

Analyzed Parameter	Mean Error (SD)	Abs Mean Error	r	*p*	¥
Contraction time (s)	−0.016 (0.045)	0.036	0.902	0.001	0.458
Positive impulse Phase time (s)	0.035 (0.082)	0.055	0.230	0.221	0.088
Negative impulse Phase time (s)	−0.051 (0.074)	0.065	0.773	0.001	0.063
Flight time (s)	0.006 (0.030)	0.024	0.737	0.001	0.599
Jump height (m)	0.009 (0.049)	0.040	0.708	0.001	0.601
RSI modified (m/s)	0.028 (0.095)	0.077	0.725	0.001	0.523
Maximum force (N)	280 (264)	331	0.491	0.006	<0.001
Minimum force (N)	−13 (96)	76	0.758	0.001	0.099

r, correlation between IMU and FP; *p*, *p*-value of the correlation between IMU and FP; ¥, *p*-value of the comparisons analysis between IMU and FP.

## Data Availability

Not applicable.

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
