# Peer review of "Accuracy of Inertial Measurement Units When Applied to the Countermovement Jump of Track and Field Athletes"

_sensors, 2022, doi:10.3390/s22197186_

Round 1

Reviewer 1 Report

The article aims to assess the countermovement jump (CMJ) with a developer encompassing the Inertial Measurement Unit. The introduction part, methods, and results are written very well. 

In the methods, please clarify the sample. Inclusion criteria etc. Did you calculate the sample size? Why did you choose six males in only two women and so on? Please write more in detail. 

In the discussion (lines 246 to 261), explain the flight phase with the IMU device in more detail. It's not so clear to understand. 

Author Response

Thank you all for this opportunity and for sharing your experience and know-how. Our team appreciated all your commentaries and suggestions and considered them essential to improve the manuscript and our knowledge.

In general, we have three different grades for English writing; however, the team considers using the Editor purpose and our manuscript has the contributions of the MDPI Language Editing Services. Regarding manuscript structure, we have, also three different grades. We improved each section with specific revisor commentaries.

About your commentaries, we appreciate your question about the sample size allowing us to clarify our difficulties on this point. We invited 30 athletes by e-mail explaining the study purposes, but only eight athletes answered and accepted our invitation. We have already clarified this point in our manuscript. About flight time, we agree with your suggestion and reinforced this point.

Thank you again, best regards 

Reviewer 2 Report

Paper carries scientific merit but the following revisions are required;

-Double check symbols in the abstract

-Ensure that all acronyms are defined

-Aims and novelty of study should be explicitly listed/stated

-Fig 3 needs improvement

-Overall flow and cohesion of paper requires improvement

-Conclusion requires strong expansions alongside a pathway for further work

-Refs section needs further inclusion of literature

Author Response

Thank you all for this opportunity and for sharing your experience and know-how. Our team appreciated all your commentaries and suggestions and considered them essential to improve the manuscript and our knowledge.

In general, we have three different grades for English writing; however, the team considers using the Editor purpose and our manuscript has the contributions of the MDPI Language Editing Services. Regarding manuscript structure, we have, also three different grades. We improved each section with specific revisor commentaries.

About your commentary, we appreciate your questions. The symbols are checked the in all manuscripts and listed the objectives making our study clearer. We changed Figure 3 to improve the font size and quality. Thank you for your suggestion about the conclusion we wrote about future approaches. Lastly, we included more than five references to reinforce some points that were not well supported.

Reviewer 3 Report

General Comments:

The study adds to a body of literature regarding appropriate devices to assess countermovement jump performance in athletes in other populations. The metrics analyzed in the present study are consistent with those of interest to the sport science community. Thus the findings are valuable to share with practitioners and researchers alike. The technical aspects of the paper were appropriate, in my perspective; however, a thorough proofreading for grammar is advised.

Specific Comments

Abstract

Line 18 – what is meant by the signal? Is it better to say “…analyze performance during the contraction phase…”

Introduction

Line 37-38 – the end of this sentence is a fragment. Please revise.

Line 43 – can removed the period after transport

Line 48 – can change to “…ease of transportation…”

Line 52 – change ‘as’ to ‘are’

Line 54-56 – can you please elaborate on evidence supporting the 5th lumbar vertebra for ideal position of IMU

Line 66 – should cite some of the many studies you mention

Line 66-69 – please provide citations to points made in this sentence

Line 69-70 – it’s not clear to me what is meant by a “encompassing IMU”. Can you please clarify for my understanding.

Line 74 – I believe you mean to say this was another hypothesis – “Additionally, we hypothesized…”

Methods

Line 79 – ‘were’ appears to be a typo

Line 82 -  for the males – how many were pole vaulters? How many decathletes?

Line 107 – please fix grammar to start the sentence

The technical and statistical approach was sound.

Results

Paragraph 1 – much of what is in text is redundant to the table 1. My preference would be to refer readers to table 1 for descriptive statistics and use text primarily to state when statistical differences were observed between the IMU and forceplate metrics.

Accuracy data paragraph, lines 189-212: same comment as for paragraph 1.

Discussion

Line 234-235 – in regards to the VAF are there cutoffs to describe the relationship between the signals established in the literature? Without more background I am challenged to interpret the meaning of the mean VAF of 64.75% provided in the results. Can you please provide more background such that readers can interpret whether this is good or bad or ?.

Line 274 – by smaller results to you mean lower values of the metrics? Supporting that the ability level of your sample was less than that of Beattie et al.?

Author Response

Thank you all for this opportunity and for sharing your experience and know-how. Our team appreciated all your commentaries and suggestions and considered them essential to improve the manuscript and our knowledge.

In general, we have three different grades for English writing; however, the team considers using the Editor purpose and our manuscript has the contributions of the MDPI Language Editing Services. Regarding manuscript structure, we have, also three different grades. We improved each section with specific revisor commentaries.

About your commentary, we appreciate your suggestion of using "signal" because we had the same doubts. We included more studies about the 5th lumbar vertebra position to reinforce the point. Regarding results, in our first version, we inserted only the tables, but when we did our last revision, we considered all results. Namely, the non-significant results, because when compared with an FP, we tried to show that IMU obtained similar results. In the discussion about VAF, we included more studies to reinforce our results. Lastly, we changed smaller to lower. As our sample aggregated male and female athletes and we have several Athletics disciplines, when compared our results with World Class male sprinters are expected to have lower values because “World-class elite male sprinters need a high elastic-explosive capacity to perform a World-Class performance.”

Round 2

Reviewer 2 Report

Thank you for making the relevant revisions